# Diversity, Abundance and Host Blood Meal Analysis of *Culicoides* Latreille (Diptera: Ceratopogonidae) from Cattle Pens in Different Land Use Types from Thailand

**DOI:** 10.3390/insects14070574

**Published:** 2023-06-22

**Authors:** Bhuvadol Gomontean, Kotchaphon Vaisusuk, Wasupon Chatan, Komgrit Wongpakam, Papasara Sankul, Laksika Lachanthuek, Ronnalit Mintara, Isara Thanee, Pairot Pramual

**Affiliations:** 1Department of Biology, Faculty of Science, Mahasarakham University, Maha Sarakham 44150, Thailand; bhuvadol.g@msu.ac.th (B.G.); may.papasara@gmail.com (P.S.); laksikalachanthuek@gmail.com (L.L.); ronnalitmintara@gmail.com (R.M.); isara.th@msu.ac.th (I.T.); 2Department of Veterinary Technology and Veterinary Nursing, Faculty of Agricultural Technology, Rajabhat Maha Sarakham University, Maha Sarakham 44000, Thailand; kotchaphon.tik@gmail.com; 3Department of Veterinary Clinic, Faculty of Veterinary Sciences, Mahasarakham University, Maha Sarakham 44000, Thailand; wasuponchatan@gmail.com; 4Walai Rukhavej Botanical Research Institute, Mahasarakham University, Maha Sarakham 44150, Thailand; komwongpa@gmail.com

**Keywords:** cattle, host blood meal, insect vector, DNA barcode

## Abstract

**Simple Summary:**

Biting midges are small blood-sucking insects that are pests and vectors of several pathogens transmitted to humans and other animals. Understanding the factors that are associated with species diversity, distribution, and abundance, as well as host preference, is crucial for the control and management of these insects. In this study, we examined species diversity, abundance, and identification of the host blood source of the biting midge genus *Culicoides* Latreille collected from cattle pens located in three different land use areas. We found that cattle pens located in villages and agricultural lands had greater abundance but were less diverse than those located close to the forest. The identification of the host blood source revealed that biting midge species in the cattle pens preferred buffalos rather than cows or chickens, although the latter two were much more abundant. Information gathered from this study will be useful for monitoring and understanding vectorial capacity as well as disease epidemiology involving the biting midge species found in cattle pens.

**Abstract:**

Biting midges of the genus *Culicoides* Latreille are significant pests and vectors that transmit pathogens to humans and other animals. Cattle are among the important livestock that can potentially be severely affected by *Culicoides*. In this study, we examined the species diversity, abundance, and host blood meal identification of biting midges in cattle pens located in three different land use types: villages, agricultural areas, and the forest edge. A total of 12,916 biting midges were collected, and most of these were from cattle pens located in villages (34%) and agricultural land (52%). Morphological identification revealed 29 *Culicoides* species. The most common species were *C. oxystoma*, *C. mahasarakhamense*, *C. peregrinus*, and *C. shortti*; taken together, these species represented >80% of all specimens collected. Despite midges being less numerous (14% of the total collection), cattle pens located near the forest showed greater diversity (23) than those from villages and agricultural areas. More diverse immature habitats and host blood sources from wildlife in nearby forests possibly explain the greater diversity in the cattle pens near the forest edge. Host blood meal analysis revealed that most (65%) biting midges had fed on buffalo despite the fact that this animal was much less numerous than cows or chickens. Relatively larger size and black-colored skin could be factors that make buffalo more attractive to biting midges than other host species. In this study, we also provided 67 DNA barcoding sequences of 13 species, three of which (*C. flaviscutatus*, *C. geminus*, and *C. suzukii*) were first reported from Thai specimens. DNA barcode analysis indicated cryptic diversity within *C. hegneri* and *C. flavescens* in Thailand, and thus, further investigation is required to resolve their species status.

## 1. Introduction

There are 1347 extant known species of *Culicoides* Latreille (Diptera: Ceratopogonidae) recorded globally [1]. Many biting midge species of the genus *Culicoides* are significant pests and vectors transmitting pathogens to humans and other animals. The important diseases caused by pathogens for which *Culicoides* can act as a vector include Oropouche fever in humans, bluetongue disease in ruminants, African horse sickness in equines, and leucocytozoonosis in chickens [2]. Outbreaks of these diseases, particularly those involving economically important livestock, can cause severe economic damage. It has been estimated that a loss of USD >1.4 billion resulted from a bluetongue virus outbreak in France [3]. The annual economic impact of African horse sickness is more than USD 95 million [4].

In Thailand, 101 species of *Culicoides* have been recorded [5,6]. However, more species certainly await discovery or formal description, as molecular genetic analysis has revealed cryptic genetic diversity in many species [7,8]. There are at least eight species of *Culicoides* recorded in Thailand that are known to be vectors of pathogens, namely, *C. brevitarsis* Kieffer, *C. fulvus* Sen and Das Gupta, *C. imicola* Kieffer, *C. peregrinus* Kieffer, *C. arakawae* Arakawa, *C. guttifer* Meijere, and *C. histrio* Johannsen, *C. oxystoma* Kieffer [2]. In addition, studies in Thailand have detected several parasitic disease agents in the following biting midge species in Thailand: *Leucocytozoon* sp., *Plasmodium juxtanucleare*, *P. gallinaceum*, *Leishmania martiniquensis* in *C. mahasarakhamense* Pramual, Jomkumsing, Piraonapicha and Jumpato [9,10,11], bluetongue virus in *C. orientalis* Macfie, *C. imicola*, *C. oxystoma* and *C. fulvus* [12], and *Trypanosoma* sp. in *C. huffi* Causey [10]. Therefore, several biting midge species in Thailand possibly have significant roles in transmitting parasitic disease-causing agents to humans as well as other animals.

Cattle are among the major host blood sources for biting midges and thus might be severely affected by diseases transmitted by these insects. In Thailand, several insect-borne diseases have caused outbreaks in cattle, including trypanosomiasis, bovine ephemeral fever, and lumpy skin disease [13,14,15]. There are >10 million cattle (beef cattle and buffalo) in Thailand, most (>60%) of which are located in the northeastern region (Department of Livestock Development, https://ict.dld.go.th/ (accessed on 26 May 2023)), where they are mostly kept by smallholder farmers, with an average of <10 cattle/farmer [16,17]. Traditionally, cattle are usually kept in temporary or semi-permanent housing (i.e., cattle pens) in the backyard near the house. Increased human population density is forcing the movement of cattle pens to more distant places, such as on agricultural land. In some cases, the cattle pens in agricultural land are located very close to natural forests. Differences in land use types, i.e., villages, agricultural, and forest edges, provide different immature habitat conditions for different biting midge species [18]. Therefore, we hypothesize that biting midge species composition and abundance in relation to the cattle pens located in different land use types will be different because of the differences in the immature stage habitats. Hence, in this study, we examined the diversity and abundance of *Culicoides* biting midge in cattle pens located in three different land use areas, i.e., within the village, in the agricultural land, and at the forest edge. We also used molecular approaches to identify the host blood sources. Additionally, the DNA barcoding sequences for species not yet reported in Thailand and species for which cryptic diversity had previously been recorded were also provided to assist species identification. This information will be very useful for monitoring and determining vectorial capacity as well as the disease epidemiology of *Culicoides* associated with cattle.

## 2. Materials and Methods

### 2.1. Specimens Collection and Identification

A total of 25 collections were made from 16 cattle pens in Thailand between January 2021 and November 2022 (Table 1 and Figure 1). These 16 sampling sites were classified into three categories based on the location of the cattle pens as follows: (i) urban, cattle pens located within the village, (ii) agricultural, cattle pens located in the agricultural area but not close (>1 km) to the natural forest, and (iii) the forest edge, cattle pens located close (<1 km) to the natural forest area. There are many techniques available for collecting *Culicoides*, although the most preferable is the use of light traps. In this study, we used sweep netting because a comparative study on animal farms has shown that light trapping and sweep netting did not yield results that were significantly different regarding abundance and species composition [19]. However, our previous studies have shown that a much greater proportion of biting midges are active before sunset [6,9], when a light trap would be less effective for collection [20]. The sweep nets used had a hoop diameter of 39 cm and a three-part telescopic handle with a total extended length of 120 cm. Four people were used to make all collections, each sweeping the net in a figure-eight motion around the animals and randomly in the air close to the cattle pens. Each person made a total of 400 sweeps between 17:00 and 19:00 p.m. at heights of approximately 0.5 m to 2.0 m from the ground. This sampling procedure was used in all collections. The specimens were preserved in 80% ethanol and stored at −20 °C until use. Specimen identification followed the keys and descriptions of Wirth and Hubert [21] and the wing pictorial illustration of Dyce et al. [22]. In addition, DNA barcoding was also used to support morphological identification.

### 2.2. Molecular Study

A total of 79 specimens were used in the COI barcoding sequence analysis. Among these specimens, 36 specimens representing seven species (*C. brevipalpis*, *C. clavipalpis*, *C. flavescens*, *C. hegneri*, *C. innoxius*, *C. palpifer* and *C. parahumeralis*) were chosen because more than one BINs were identified in BOLDs (https://www.boldsystems.org/index.php) (accessed on 26 May 2023) for these nominal species. Therefore, COI barcoding sequences should allow us to identify the BIN that the specimens obtained in the present study belong to. The remaining specimens (*n* = 43) from six species (*C. flaviscutatus*, *C. geminus*, *C. suzukii*, *Culicoides* sp. 1, 2, and 3) were selected because their representative COI sequences have not yet been reported from Thailand. DNA was extracted from the whole individual using the GF-1 Nucleic Acid Extraction Kit (Vivantis Technologies Sdn. Bhd, Shah Alam, Malaysia). Polymerase chain reaction (PCR) was used to amplify the COI gene barcoding region using the primers LCO1490 (5′-GGTCAACAAATCATAAAGATATTGG-3′) and HCO2198 (5′-TAAACTTCAGGG TGACCAAAAAATCA-3′) [23]. The PCR reaction conditions followed those used by Takawanit et al. [24]. PCR products were checked with 1% agarose gel electrophoresis and purified using a PureDireX PCR CleanUp & Gel Extraction Kit (Bio-Helix, Taiwan, China).

Host blood meal identification was based on the mitochondrial cytochrome b (cyt *b*) gene sequence. Adult females were checked for host blood in their abdomen under a microscope. Representative blood-fed specimens of every species collected were chosen for blood meal source identification, except for eight species (*C. albibasis*, *C. homotomus*, *C*. nr. *shortti*, *C. suzukii*, *C*. (*Trithecoides*) sp., *Culicoides* sp. 1, 2 and 3) for which we did not find blood-fed females in our collections. In total, 141 blood-fed specimens from 21 species were used for the molecular identification of the host blood source. DNA was extracted from the whole individual using the same DNA extraction kit as used for the COI gene. The mitochondrial cyt *b* of the vertebrate host blood in the blood meal was amplified using the primers L14841 and H15149 [25]. The PCR reaction conditions and temperature profiles for the cyt *b* amplification followed the method described in Malmqvist et al. [26]. Both COI and cyt *b* PCR products were sent for DNA sequencing at ATCG Company Limited (Thailand Science Park (TSP), Pathumthani, Thailand) using the same primers as used for PCR.

### 2.3. Data Analysis

The diversity and abundance parameters, namely, relative abundance (RA), occurrence (OC), and species richness (S) were calculated for each collection. For the relative abundance, RA = ni/N × 100, ni is the number of specimens of species, i, and N is the total number of specimens collected. Species richness, S, is the total number of species in each collection. The occurrence (OC) is the ratio of the number of collections where species occur and the total number of collections [27]. This parameter, according to Gherbi et al. [28], enables patterns of *Culicoides* species distribution to be classified into five categories, as follows: (i) sporadic (OC ≤ 20%), (ii) infrequent (OC > 20% but ≤40%), moderate (OC > 40% but ≤60%), frequent (OC > 60% but ≤80%), and constant (OC > 80%). Two diversity indices, Shannon *H*’ index and evenness (E), were estimated for the three habitat types using PAST version 1.81 [29]. The differences in *Culicoides* species assemblages between the cattle pens in urban, agricultural, and forest edge areas were tested using analysis of similarities (ANOSIM) [30]. ANOSIM analysis was performed using PAST version 1.81 [29].

To identify species based on the COI DNA barcoding sequence, the identification system in BOLD (https://www.boldsystems.org/index.php/IDS_OpenIdEngine) (accessed on 26 May 2023) [31] was used. Intraspecific genetic divergence based on the p-distance was calculated in TaxonDNA [32]. The genetic relationships between 13 nominal species based on the 324 COI sequences (67 obtained in this study, and the remaining were retrieved from BOLD, https://www.boldsystems.org/index.php) (accessed on 26 May 2023) were inferred based on the Kimura 2-paramter in MEGA X [33]. Branch support was calculated via bootstrapping with 1000 replications. Sequences of black fly, *Simulium chumpornense* (MT262569–70), were used as the outgroup. FigTree v1.4.3 (http://tree.bio.ed.ac.uk/software/figtree/) (accessed on 26 May 2023) was used to visualize and prepare graphics of the NJ tree. To determine the source of the blood meal of *Culicoides*, the cyt *b* sequences were compared to those reported in GenBank using the Basic Local Alignment Search Tool (BLAST) (https://blast.ncbi.nlm.nih.gov/Blast.cgi) (accessed on 26 May 2023).

## 3. Results

### 3.1. Species Diversity and Abundance of Culicoides in Cattle Pens in Three Habitat Types

A total of 12,916 adult specimens belonging to 29 species of *Culicoides* biting midge were collected across 25 collections from 16 locations in Thailand (Table 2, Figure 2). Species richness in each collection ranged between 3 and 18 (Table 1), and 15 of the 25 collections found >10 species. The sampling sites that possessed the greatest species diversity (18 species) were NK1 in Nong Khai province and UB2 in Ubon Ratchathani province, both in northeastern Thailand, and were located close to the Mekong River. The most abundant species occurring in animal shelters was *C. oxystoma*. This species represented more than 44% (5706 of 12,916) of the biting midge specimens collected. Three other species also had relatively high abundance, namely, *C. mahasarakhamense* (RA = 17.2%), *C. peregrinus* (RA = 10.8%), and *C. shortti* (RA = 8.16%). The remaining species had low abundance (RA < 5%). Four species were considered to be constant (OC ≥ 80%), *C. mahasarakhamense* (OC = 92%), *C. oxystoma* (OC = 84%), *C. peregrinus* (OC = 84%), and *C. actoni* (OC = 80%) (Table 2). *Culicoides* specimens were mostly collected (6775, 52.45%) from cattle pens located in the agricultural areas, followed by those from the village areas (4360, 33.76%). Cattle pens in the forest edge had the least abundance, with only 1781 (13.79%) specimens collected.

The numbers of *Culicoides* species collected from the cattle pens located in the three habitat types were 21, 18, and 22 for urban, agricultural, and forest edge, respectively (Table 2). One-way ANOVA analysis revealed no statistically significant difference in terms of species richness in these three areas (*F* = 0.413, *p* = 0.666, d.f. = 2). However, the Shannon *H’* index indicated that cattle pens located at the forest edge area possessed greater *Culicoides* diversity (*H’* = 2.37) than the urban (*H’* = 1.80) and the agricultural (*H’* = 1.53) areas. Cattle pens near the forest edge also had the highest *E* value (*E* = 0.75) compared to urban (*E* = 0.59) and agricultural (*E* = 0.54) areas. This indicated that *Culicoides* species occurring in the cattle pens located close to the forest were more similar in abundance compared to those of agricultural and urban areas. ANOSIM analysis revealed significant differences between the *Culicoides* community at the forest edge compared to those in urban (*R* = 0.1757, *p* = 0.0437) and agricultural areas (*R* = 0.3961, *p* = 0.0026), but the latter two areas were not significantly different from each other (*R* = 0.0678, *p* = 0.1579).

### 3.2. DNA Barcoding and Phylogenetic Tree

A total of 67 COI sequences (GenBank accession nos. OR073892–921, OR073923–30, OR073944–49, OR073957–65, OR073974–79, and OR074028–37) representing 13 species were obtained in this study although 79 specimens were subjected to DNA analysis. Three species, *C. flaviscutatus* Wirth and Hubert, *C. geminus* Macfie, and *C. suzukii* Kitaoka, were reported for the first time as being collected in Thailand. The maximum intraspecific genetic divergence among these species varied between 0.16% in *C. flavescens* and *Culicoides* sp. 1 and 3.00% in *C. geminus* (Table 3). The identification of these specimens using the BOLD identification engine (https://www.boldsystems.org/index.php/IDS_OpenIdEngine) (accessed on 26 May 2023) found that only 50.7% (34 from 67) of seven species were successfully identified (*C. brevipalpis* Ingram and Macfie, *C. flavescens* Kieffer, *C. flaviscutatus* (five, successful; three, no match), *C. geminus*, *C. hegneri* Causey, *C. innoxius* Sen and Das Gupta, *C. parahumeralis* Wirth and Hubert (one, successful; one, ambiguous). Twenty-five specimens from two nominal species, *C. clavipalpis* Mukerji and *C. suzukii*, and three unidentified species, *Culicoides* sp. 1–3, had no sequence match at the species level in BOLDs. Specimens (7) morphologically identified as *C. palpifer* Das Gupta and Ghosh were ambiguously identified in BOLDs, which classified them as *C. parahumeralis*/*C. palpifer* (Table 3).

The neighbor-joining tree based on 324 COI sequences (67 from this study) of 13 species revealed that all were monophyletic with strong support (>95%) (Figure 3). The only exceptions were *C. palpifer* and *C. parahumeralis*, as both species were clustered together, forming a polyphyletic clade. Deeply divergent clades existed in six nominal species. Nine divergent subclades were found in *C. innoxius*. Six specimens obtained in this study belonged to a subclade of the BIN BOLD: ADV2561, which includes specimens from Thailand, Malaysia, and China. In this study, *Culicoides brevipalpis* (*n* = 2) formed a subclade with members of BIN BOLD: AAJ7389 that had been collected from Japan, Timor-Leste, Papua New Guinea, China, Bangladesh, and Thailand. A specimen of *C. hegneri* obtained in the present study clustered with the sequence of this species collected from Thailand. These specimens formed a different clade from the two other BINS of this species, and both are from India. In this study, two specimens of *C. parahumeralis* belonged to a clade representing BIN BOLD: ADK3441, together with those from Malaysia, China, Timor-Leste, and Thailand. *Culicoides palpifer* (*n* = 7) obtained in this study formed a clade with members of BIN BOLD: ADT9601 from Malaysia, Philippines, China, Vietnam, and Thailand.

### 3.3. Host Blood Meal Analysis

A total of 2935 blood-engorged females were collected, and most of these (2208, or 75%) were specimens of *C. oxystoma* (1651) and *C. peregrinus* (557). From a total of 141 blood-engorged female specimens subjected to molecular identification of host blood source, only 77 specimens (54.6%) from 16 species were successful in terms of the amplification of the vertebrate blood DNA (Figure 4). Fifty specimens were identified as having blood from buffalo (*Bubalus bubalis*), with a sequence similarity of the cyt *b* gene ranging between 98 and 100%. The following *Culicoides* species were found to feed on buffalo: *C. actoni* (*n* = 6), *C. brevipalpis* (*n* = 7), *C. brevitarsis* (*n* = 3), *C. flaviscutatus* (*n* = 8), *C. geminus* (*n* = 4), *C. parahumeralis* (*n* = 3), *C. imicola* (*n* = 3), *C. hegneri* (*n* = 1), *C. oxystoma* (*n* = 4), *C. palpifer* (*n* = 6), and *C. shortti* (*n* = 3). Ten specimens were identified as having blood from cows (*Bos indicus*/*Bos taurus*), with a sequence similarity ranging between 98 and 100%. The *Culicoides* species that feed on cows were *C. oxystoma* (*n* = 1), *C. shortti* (*n* = 2), and *C. innoxius* (*n* = 7). Fourteen specimens were identified as having blood from chicken (*Gallus gallus*). *Culicoides* species found to feed on chicken were *C. clavipalpis* (*n* = 1), *C. guttifer* (*n* = 5), *C. parahumeralis* (*n* = 5), and *C. mahasarakhamense* (*n* = 3). Blood from two specimens (one from *C. brevitarsis* and one from *C. flaviscutatus*) was identified as belonging to red collared dove (*Streptopelia tranquebarica* Hermann). *Culicoides brevitarsis* was also found to feed on humans (*n* = 1).

## 4. Discussion

### 4.1. DNA Barcoding

The morphological identification of *Culicoides* can be problematic because of their small size (<2.5 mm in body length) [2] and the limitations of the diagnostic morphological characters, particularly in terms of distinguishing between closely related species [34]. Therefore, it is preferable to use additional species identification tools, such as DNA barcodes, to support morphological identification [34]. One of the major obstacles concerning the use of DNA barcodes is a lack of reference sequences in the databases. In Thailand, at least one COI barcoding sequence is available for 59 out of 101 *Culicoides* species recorded in the country [8]. The small sample size and narrow geographical range of some species made representative sampling unlikely. Thus, future work should deploy significant efforts to increase the sample size collected across the species’ geographic range to address the range of genetic diversity [8].

In this study, we obtained 67 COI barcoding sequences from 13 nominal species of *Culicoides* in Thailand. Among these species, three (*C. flaviscutatus*, *C. geminus,* and *C. suzukii*) were the first reported examples found in Thailand. Among the 67 sequences, for 37 (55%) there was agreement between the morphological and BOLD COI identifications. The remaining specimens that did not agree with the morphological identification arose because of the ambiguous identification of *C. palpifer* and *C. parahumaeralis*, and individuals could not be identified because of the lack of reference sequences in BOLDs for *C. suzukii*, *Culicoides* sp. 1, 2, and 3. Of the six species for which DNA barcodes and morphology both supported species identification, three (*C. innoxius*, *C. hegneri*, and *C. brevipalpis*) had >1 BINs and showed deeply genetically divergent clades in the NJ tree. *Culicoides innoxius* had nine BINs identified in BOLDs corresponding with the divergent clades in the NJ tree. The type locality of this species is in India [21]; thus, presumably, the sequences of this species reported from that country are more likely to be true *C. innoxius*. The sequences of species morphologically identified as *C. innoxius* in the present study belonged to a BIN that included specimens reported from China, Thailand, and Malaysia. This BIN was 4.05% different from Indian specimens, suggesting that they are different species, and it is considered that further morphological characters plus additional genetic markers are required to investigate. A similar situation was also found in *C. hegneri,* where three genetically divergent lineages were revealed. The type locality of this species is in Chiang Rai province, northern Thailand [21]. Two sequences from Thailand, one obtained in the present study, which was collected from the northeastern region, and another collected from Trang province, southern region, by Gopurenko et al. [8], belong to the same clade but with >5% genetic differentiation. Therefore, these two specimens could be different species. According to Wirth and Hubert [21], the immature stage of this species is presumed to prefer mud at the margins of large streams or rivers. Specimens of this species obtained in the present study were collected from location (UB2), close to the Mekong River, a location which agrees with the presumed preferred habitat of the immature stage of the species. Two other clades, both from India, were also genetically highly (>10%) different from Thailand’s *C. hegneri*. Further investigation is needed to clarify the species status of this species. Two genetic lineages in *C. brevipalpis* have been reported previously [35]. The sequences obtained in the present study belong to the BIN, with members of specimens from Thailand previously reported [7,8]. This BIN also included specimens from other countries, namely Japan, Timor-Leste, Papua New Guinea, China, and Bangladesh.

### 4.2. Species Diversity and Abundance of Culicoides in Cattle Pens

Among 12,916 adult flies collected from cattle pens, four species, *C. oxystoma*, *C. mahasarakhamense*, *C. peregrinus,* and *C. actoni* were distributed widely, with >80% occurrence in the sampling sites. This result agrees with previous information that they are well-adapted to diverse environmental conditions [21]. The first three species were also very abundant, representing >72% of all specimens collected. The results were consistent with other studies conducted in Thailand that found that these species were very abundant and commonly found in diverse habitats [10,12,36]. Studies in other countries also found that *C. oxystoma* and *C. peregrinus* were among the most common species observed in livestock farms [37,38,39].

Diversity and abundance of *Culicoides* species are related to biotic and abiotic environmental factors, such as climatic conditions, the suitability of immature habitats, host types, and host density [40,41]. In this study, we collected biting midge specimens from cattle pens; thus, the sampling sites did not differ in terms of host availability, except for those at the forest edge where additional wildlife animals can potentially be hosts. All sampling sites, except for the one in the south, were located in the northeastern region and experienced similar climatic conditions. Therefore, differences in the species diversity and assemblages of the *Culicoides* in the three habitat types were most likely related to the habitat conditions experienced by the immature stages of *Culicoides* and by the availability of these habitat types. We found that cattle pens located close to the forest had greater species richness, although this difference was not statistically significant. Twenty-two species from a total of twenty-nine (76%) were found in the cattle pens located near the forest. The forest edge sites also had a greater Shannon *H*’ index and evenness value than the urban and agricultural sites. The species assemblage of the forest edge was also significantly different from those of urban and agricultural sites. Seven species (*C. boophagus*, *C. clavipalpis*, *C. flavescens*, *C. hegneri*, *C*. nr. *shortti*, *Culicoides* sp. 2, and 3) were exclusively associated with forest edge habitat. Species distributions and abundances associated with the forest area have been reported in other regions, such as South America [42] and Europe [43]. Because both immature habitat and host blood source are related to local species diversity and abundance [41], the greater diversity of *Culicoides* in the cattle pens close to the forest found in the present study is, therefore, most likely due to the more diverse immature habitats and animal hosts in the forest than in the urban (village) and agricultural lands [44].

### 4.3. Host Blood Meal Identification

Information on the host blood meal is important in the determination of the vectorial capacity and disease epidemiology of hematophagous insect species [45,46]. In Thailand, host blood meal sources have been reported for 21 biting midge species based on molecular identification [7,10,36,47]. In this study, we report additional host blood sources for 16 species in Thailand. Among these species, *C. hegneri* (feeds on buffalo), *C. brevipalpis* (feeds on buffalo), *C. parahumeralis* (feeds on buffalo and chicken), *C. clavipalpis* (feeds on chicken), and *C. palpifer* (feeds on buffalo) were the first reports of the host blood sources in Thailand. The identification of host blood sources of *C. hegneri* in India also found that this species feeds on cattle (*Bos indicus*/*Bos taurus*) and that *C. palpifer* feeds on buffalo [48].

The most important (65%) host blood animal detected in this study was buffalo. A high number of species (10) and a high number of females (50) that favor buffalo in terms of feeding rather than other available hosts, such as cows, humans and chickens, were found in the areas where specimens were collected. The results are different from those of a recent study conducted in Thailand, as the majority (68%) of *Culicoides* blood meals were derived from mixed sources (cows and birds) [36]. This disparity arises due to the utilization of conventional PCR in our study for the amplification of the vertebrate cyt *b* gene, which was, consequently, unable to identify the mixed-host blood meals in an individual specimen. In contrast, Sunantaraporn et al. [36] employed multiplex PCR, enabling them to identify the mixed-host blood in an individual specimen.

A similar situation where buffalo is the preferred blood source for *Culicoides* has also been reported in India [48]. The greater tendency to feed on buffalo despite their being present in lower numbers than cattle possibly relates to their larger size; therefore, they have a larger surface area for midges to land on. In addition, buffalo have black skin, possibly making it preferable to the white or brown skin of cows. An experimental study found that *C. impunctus* had a greater preference for black than white color stripes [49]. Furthermore, buffalo also have less hair to protect the skin compared to cows or chickens. These factors could enhance biting midge preferences for feeding on buffalo.

## 5. Conclusions

In this study, we found that biting midges were more abundant in cattle pens located in agricultural areas rather than urban areas. However, the high abundance in these areas predominately involved a few common species, such as *C. oxystoma*, *C. peregrinus*, *C. shortti,* and *C. mahasarakhamense,* which are known to be associated with cattle (*C. oxystoma*, *C. shortti*, *C. peregrinus*) and domestic chickens (*C. mahasarakhamense*). The cattle pens located close to the forest showed the greatest species diversity index and there were seven species unique to this habitat type. This was possibly due to the more diverse breeding sites and hosts available in the forest compared to those of village and agricultural areas. Further studies should be conducted to monitor the population dynamics of those species in forest edge locations because they might provide a bridge for pathogen transfer between wildlife and domestic animals [18]. Cattle movement relating to either grazing or trade transport might thus facilitate pathogen-carrying midges or pathogens being transferred between forest wildlife hosts and domestic animals and expedite transmission to humans. Therefore, monitoring and vector control should also focus on the cattle pens near forests.

## Figures and Tables

**Figure 1 insects-14-00574-f001:**
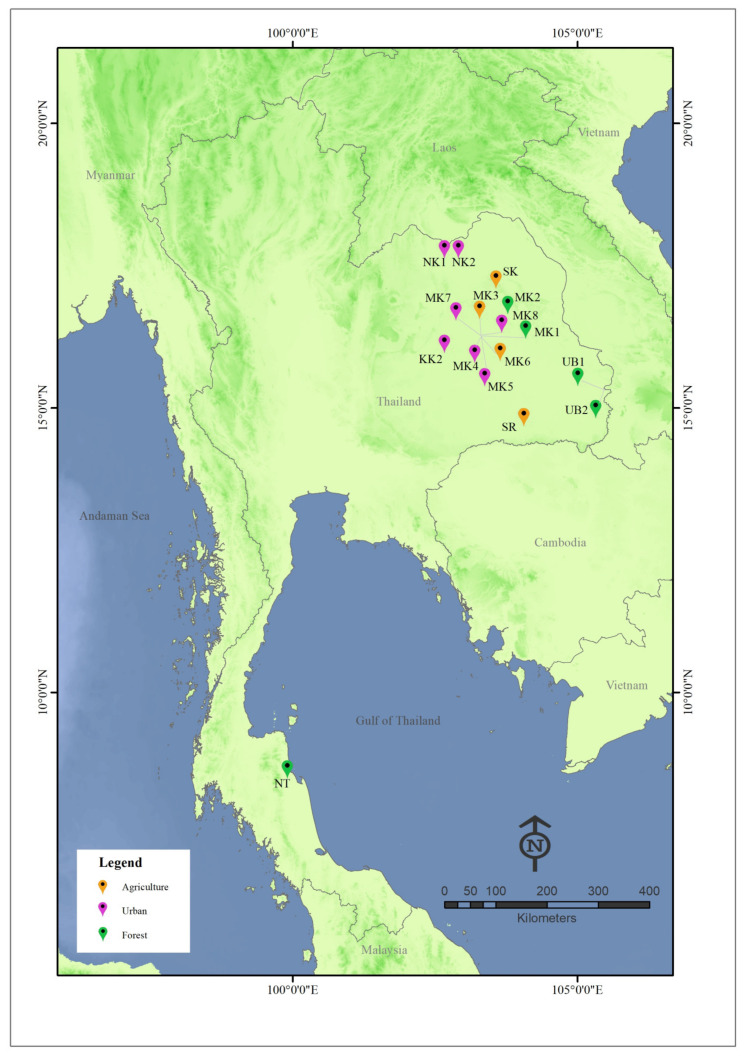
Map (modified from http://mitrearth.org (accessed on 1 August 2022)) indicating the 16 sampling locations of *Culicoides* from Thailand used in this study. Details of sampling locations are included in Table 1. Locality symbols are labeled according to the habitat types.

**Figure 2 insects-14-00574-f002:**
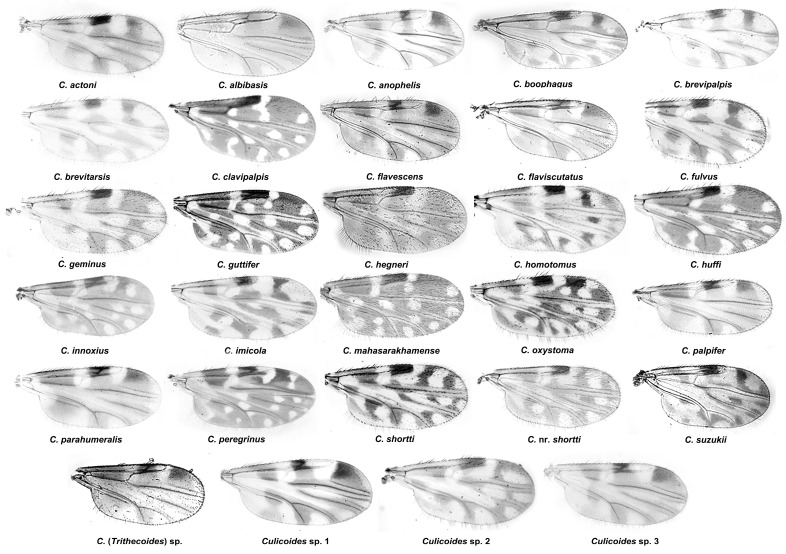
Photographs of female wings of 29 species of *Culicoides* collected from cattle pens in Thailand.

**Figure 3 insects-14-00574-f003:**
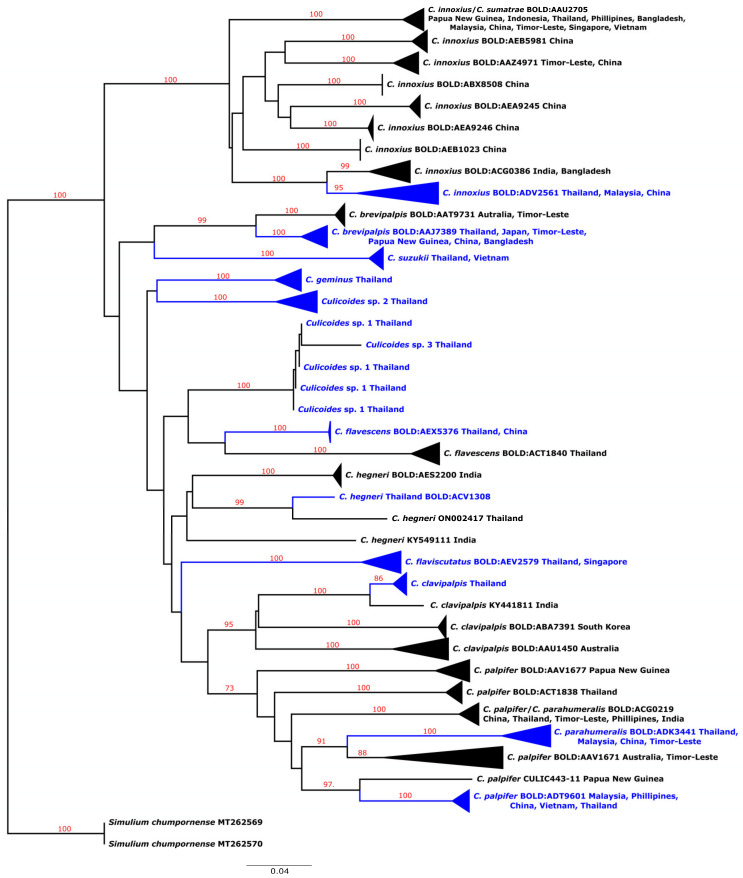
Neighbor-joining tree based on 324 COI sequences of 13 species of the *Culicoides* obtained in this study (*n* = 67, blue) and those publicly reported in BOLDs (black) (https://www.boldsystems.org/index.php) (accessed on 26 May 2023). Variability within each clade was collapsed into triangles for visualization. The Barcode Index Number (BIN) assigned by BOLDs and the countries of collection site of members within each BIN are shown after the species name. Bootstrap values are shown above the branch.

**Figure 4 insects-14-00574-f004:**
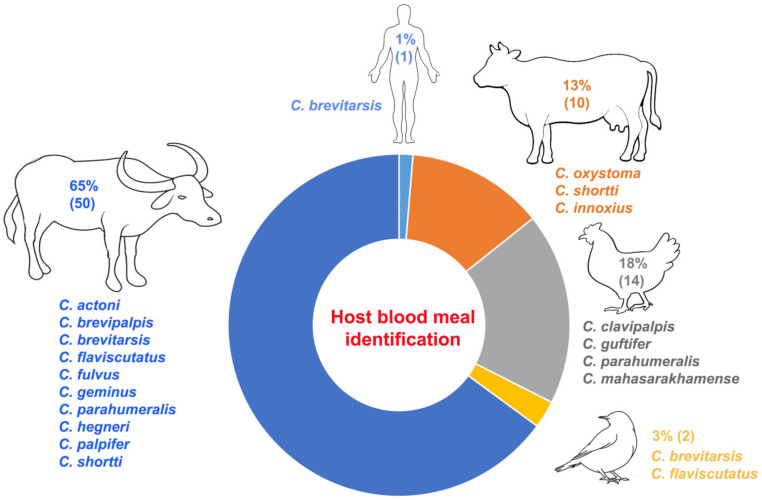
Host blood meal identifications of 77 blood-engorged females from16 biting midge species based on the cyt *b* gene sequences.

**Table 1 insects-14-00574-t001:** Sampling location and number of *Culicoides* biting midge collected from cattle pens in Thailand.

Location: Type ^a^: Code	Animal Host (n)			Date	Latitude/Longitude	S	N
	Cow	Buffalo	Chicken ^b^				
Prangku, Sisaket: A: SR1-1	19	0	>50	2 January 2021	14.83034 N/104.06043 E	8	125
A: SR1-2	19	0	>50	7 March 2021	14.83034 N/104.06043 E	10	396
A: SR1-3	19	0	>50	25 January 2022	14.83034 N/104.06043 E	11	521
A: SR1-4	19	0	>50	12 February 2022	14.83034 N/104.06043 E	11	1263
A: SR1-5	19	0	>50	13 February 2022	14.83034 N/104.06043 E	15	1109
A: SR1-6	19	0	>50	4 June 2022	14.83034 N/104.06043 E	11	802
Meuang, Maha Sarakham: F: MK1-1	12	0	>25	9 January 2021	16.23489 N/103.38491 E	7	19
F: MK1-2	12	0	>25	9 March 2021	16.23489 N/103.38491 E	11	717
Na si nuan, Kantharawichai, Maha Sarakham: F: MK2	10	0	>25	7 February 2021	16.34525 N/103.20475 E	12	243
Ban Wai (1), Kantharawichai, Maha Sarakham: A: MK3-1	2	0	>100	20 January 2021	16.30737 N/103.18887 E	3	397
A: MK3-2	2	0	>100	5 February 2021	16.30737 N/103.18887 E	5	378
Ban Wai (2), Kantharawichai, Maha Sarakham: U: MK4	10	0	>100	24 February 2021	16.30145 N/103.18396 E	5	501
Ban Don Wiangjan, Maha Sarakham: U: MK5	20	0	>25	25 March 2021	16.25617 N/103.26658 E	6	843
Ban Yang, Kantharawichai, Maha Sarakham: A: MK6-1	5	6	>100	21 January 2022	16.29041 N/103.18349 E	11	522
A: MK6-2	5	6	>100	13 February 2022	16.29041 N/103.18349 E	11	1262
Chiang Yuen, Maha Sarakham: U: MK7-1	25	3	>100	14 June 2022	16.37175 N/103.06013 E	12	138
U: MK7-2	25	3	>100	29 June 2022	16.37175 N/103.06013 E	8	798
Ban Don Suan, Maha Sarakham: U: MK8	5	3	>100	26 February 2021	16.25266 N/103.27520 E	10	252
Ban Phai, Khon Kaen: U: KK2	15	0	>100	9 July 2022	16.11994 N/102.72616 E	9	669
Waritchaphum, Sakhon Nakhon: A: SK	8	5	>50	28 March 2021	17.24221 N, 103.57464 E	5	85
Thasala, Nakhon Si Thamarat: F: NT	40	0	0	20 September 2022	8.64028 N/99.90361 E	7	140
Wiangkook (1), Nong Khai: U: NK1	15	3	>50	2 July 2022	17.78402 N/102.66318 E	18	887
Wiangkook (2), Nong Khai: U: NK2	10	8	>50	3 July 2022	17.77999 N/102.66162 E	12	272
Khong Chiam 1, Ubon Ratchathani: F: B1	2	0	>100	11 November 2022	15.31574 N/105.51204 E	12	82
Khong Chiam 2, Ubon Ratchathani: F: B2	5	0	>25	12 November 2022	15.31459 N/105.49841 E	18	495
Total						29	12,916

^a^ Type of land used: U = urban; F = forest edge; A = agriculture; S = species richness; N = number of specimens. ^b^ Exact number of chickens in the cattle pens could not be counted. The number presents an estimate based on information from the cattle pen owners.

**Table 2 insects-14-00574-t002:** List of species, number of specimens, relative abundance (RA), occurrence (OC), and host blood meal sources of *Culicoides* collected from the cattle pens in Thailand.

Species	N	RA (%)	% OC (Category) ^a^	Land Use Type ^b^	Host Blood Meal Identification (Sampling Location Code ^c^)
*C. actoni* Smith	322	2.49	80 (C)	F, U, A	buffalo (NK1, NK2, SR1-4)
*C. albibasis* Wirth and Hubert	10	0.08	8 (S)	U	
*C. anophelis* Edwards	56	0.43	36 (M)	F, U, A	
*C. boophagus* Macfie	80	0.60	4 (S)	F	
*C. brevipalpis* Delfinado	18	0.14	20 (I)	F, U, A	buffalo (MK6-1, MK8)
*C. brevitarsis* Kieffer	578	4.48	64 (F)	F, U, A	buffalo (MK8), human (MK8), bird (red collared dove) (MK8)
*C. clavipalpis* Mukerji	5	0.04	4 (S)	F	chicken (UB2)
*C. flavescens* Macfie	5	0.04	4 (S)	F	
*C. flaviscutatus* Wirth and Hubert	423	3.28	8 (S)	U	buffalo (NK1), bird (red collared dove) (NK1)
*C. fulvus* Sen and Das Gupta	108	0.84	68 (F)	F, U, A	buffalo (SR1-6)
*C. geminus* Macfie	90	0.70	20 (I)	U, A	buffalo (MK5)
*C. guttifer* de Meijere	216	1.67	72 (F)	F, U, A	chicken (UB2)
*C. hegneri* Causey	28	0.22	8 (S)	F	buffalo (UB1)
*C. homotomus* Kieffer	11	0.08	32 (I)	U, A	
*C. huffi* Causey	125	0.97	72 (F)	F, U, A	
*C. innoxius* Sen and Das Gupta	37	0.29	28 (I)	F, U, A	cow (SR1-2, UB2)
*C. imicola* Kieffer	206	1.60	72 (F)	F, U, A	buffalo (MK5)
*C. mahasarakhamense* Pramual et al.	2225	17.23	92 (C)	F, U, A	chicken (UB2)
*C. oxystoma* Kieffer	5706	44.18	84 (C)	F, U, A	buffalo (MK4), cow (UB1)
*C. palpifer* Das Gupta and Ghosh	57	0.44	12 (S)	F, U	buffalo (NK1, NK2)
*C. parahumeralis* Wirth and Hubert	25	0.19	20 (S)	F, U, A	buffalo (UB1), chicken (UB2)
*C. peregrinus* Kieffer	1398	10.83	84 (C)	F, U, A	
*C. shortti* Smith and Swaminath	1054	8.16	76 (F)	F, U, A	buffalo (SR1-5), cow (SR1-6, MK1-2)
*C.* nr. *shortti*	37	0.29	8 (S)	F	
*C. suzukii* Kitaoka	23	0.18	4 (S)	A	
*C.* (*Trithecoides*) sp.	24	0.19	12 (S)	U, A	
*Culicoides* sp. 1	33	0.30	16 (S)	U	
*Culicoides* sp. 2	14	0.11	12 (S)	F	
*Culicoides* sp. 3	2	0.02	4 (S)	F	
Total	12,916				

^a^ Occurrence category: S = sporadic; I = infrequent; M = moderate; F = frequent, C = constant [28]. ^b^ Land use type: U = urban; F = forest edge; A = agriculture; N = number of specimens. ^c^ Details of sampling location are included in Table 1.

**Table 3 insects-14-00574-t003:** DNA barcode identification of the *Culicoides* collected from cattle pens in Thailand.

Morphological Species (*n*)	Intraspecific Genetic Divergence (%)	BOLD Identification (BIN of the Specimens in the Present Study)	Nearest Neighbor BIN (min. Distance)	Nearest Neighbor BIN Species
*C*. *brevipalpis* (2)	0.66	*C. brevipalpis* (AAJ7389)	AAT9731 (6.81%)	*C*. *brevipalpis*
*C*. *clavipalpis* (5)	0.17–0.86	No match	-	-
*C*. *flavescens* (2)	0.16	*C. flavescens* (AEX5376)	ADL2471 (12.47%)	*C. fagineus*
*C*. *flaviscutatus* (8)	0–2.45	*C*. *flaviscutatus* (AEV2579)	AEU9734 (2.97%)	*Culicoides* sp.
*C*. *geminus* (19)	0–3.00	*C. geminus* (ABW1357)	ADD8604 (10.59%)	Ceratopogonidae sp.
*C*. *hegneri* (1)	N/A	*C. hegneri* (ACV1308)	AFD7233 (1.83%)	*C. hegneri*
*C*. *innoxius* (5)	0–0.48	*C*. *innoxius* (ADV2561)	ACG0386 (4.05%)	*C. innoxius*
*C*. *palpifer* (7)	0–1.73	*C*. *palpifer*/*C*. *parahumeralis* (ADT9601)	AET5852 (7.24%)	*C. palpifer*
*C*. *parahumeralis* (2)	0	*C*. *palpifer*/*C*. *parahumeralis* (ADK3441)	AEB2961 (11.05%)	*C. parahumeralis*
*C*. *suzukii* (6)	0.21–1.04	No match	-	-
*Culicoides* sp. 1 (4)	0–0.16	No match	-	-
*Culicoides* sp. 2 (5)	0.68–1.88	No match	-	-
*Culicoides* sp. 3 (1)	N/A	No match	-	-

## Data Availability

The data generated during the study have already been reported in the manuscript.

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
