# Peer review of "Diversity, Abundance and Host Blood Meal Analysis of Culicoides Latreille (Diptera: Ceratopogonidae) from Cattle Pens in Different Land Use Types from Thailand"

_insects, 2023, doi:10.3390/insects14070574_

Round 1

Reviewer 1 Report

Dear Authors

Please correct:

Material & Methods - Page 3, lines 92-93 – the authors refer 16 sampling places. However, in Fig 1 there is reference to 15 sampling places. Please check. 

Page 3, lines 112-113 – please specify if DNA extraction was done using the whole insect or just parts of the body.

Page 4 – Table 1 – in caption include which means S and N abbreviations.

Also, in Table 1 caption – replace “…of chicken in the cattle pens cannot count” with “…of chicken in the cattle pens could not be counted”. 

Page 6 – line 159 – italicize “Culicoides”.

Page 7, Table 2 – concerning C. brevitarsis, correct “red collar dove” to “red collared dove”. Also, for C. flaviscutatus, check out “red collar bird”. Do you meant “red collared bird” or “red collared dove”?

Table 2 – caption - standardise the presentation of abbreviations (see Table 1).

Page 8 – Fig 2 - some of the backgrounds impair the observation of the wing patterns. White backgrounds will work better.

Page 10, line 251 – specify if the whole specimen was used for molecular identification.

Page 10, line 255 – in “…buffalo; C. actoni…” replace the semicolon with a colon.

A well-structured manuscript clearly written.

Author Response

Reviewer 1

Comments and Suggestions for Authors

Dear Authors

Please correct:

Material & Methods - Page 3, lines 92-93 – the authors refer 16 sampling places. However, in Fig 1 there is reference to 15 sampling places. Please check.

Author response: Thank you for your comment. We have checked and confirmed that there are 16 sampling place in Figure 1. The number of sampling site in Figure 1 caption was corrected.

Page 3, lines 112-113 – please specify if DNA extraction was done using the whole insect or just parts of the body.

Author response: We used the whole individual for DNA extraction. This information has been added to the text.

Page 4 – Table 1 – in caption include which means S and N abbreviations.

Author response: Thank you for recommendation, meaning of S and N have been added.

Also, in Table 1 caption – replace “…of chicken in the cattle pens cannot count” with “…of chicken in the cattle pens could not be counted”.

Author response:  Thank you for your suggestion, we have revised this sentence accordingly.

Page 6 – line 159 – italicize “Culicoides”.

Author response: Corrected

Page 7, Table 2 – concerning C. brevitarsis, correct “red collar dove” to “red collared dove”. Also, for C. flaviscutatus, check out “red collar bird”. Do you meant “red collared bird” or “red collared dove”?

Author response: Thank you for your suggestion. The correct spelling is actually “red collared-dove”. We have corrected this typing error.

Table 2 – caption - standardise the presentation of abbreviations (see Table 1).

Author response:  Thank you for suggestion, we have revised the abbreviations to make them consistent throughout the manuscript.

Page 8 – Fig 2 - some of the backgrounds impair the observation of the wing patterns. White backgrounds will work better.

Author response: Thank you very much for your suggestion, we have modified this figure.

Page 10, line 251 – specify if the whole specimen was used for molecular identification.

Author response: Thank you for your suggestion, we have added the information of the DNA extraction for host blood meal analysis in Materials and Methods.

Page 10, line 255 – in “…buffalo; C. actoni…” replace the semicolon with a colon.

Author response:  Corrected

Comments on the Quality of English Language

A well-structured manuscript clearly written.

Author response:  Thank you very much for your comments and suggestions that are very helpful for the improvement of the manuscript.

Reviewer 2 Report

This manuscript presents interesting results of sampling Culicoides around cattle pens in Thailand, including morphological and COI identifications and blood-meal analyses.  The following concerns should be addressed:

Please describe the sweep-netting procedure more clearly.  Sweep-netting should follow a standardized procedure if comparisons are to be made among locations. For example, the number of sweeps, height of the sweeps, and their configuration (typically a figure-8 motion) should be consistent for every sample. Otherwise, comparisons among sites becomes questionable.

If 25 collections were made from 16 sites, how were the collections partitioned?  Clearly, some sites would have only one collection whereas other sites would have multiple collections.  Unless there was a standardized collecting procedure, comparisons among sites will be biased. For example, species richness would be expected to be proportionally greater where more collections were made.

If 12,916 midges were collected, how was the very small number of 79 specimens chosen for COI barcoding?

How were specimens selected for blood-meal analysis?  How many were analyzed?  (We do not learn the answer until line 251).  Were some of the same 79 specimens for COI barcoding used?  Was visible evidence of a blood meal used to determine which specimens would be used?  Please provide more explanation of all this.

Under section 2.2 Molecular Study, please separate the COI barcoding analysis and the blood-meal analysis into two separate paragraphs to avoid confusion.

Table 1:  For the table, pleased define S in a footnote.  Footnote b (“Exact number of chicken…from cattle pens owner”) does not make sense.  Please correct the English.

Figure 1.  Actually the figure shows much more than “Map of Thailand”, including other countries.  It shows a portion of Southeast Asia, including Thailand.

Figure 2 seems to be a mix of color and black and white photos. All photos should be black and white. Otherwise, the photos suggest that there is some significance to the different colors of wings.

Lines 264-265 state that C. brevipalpis fed on humans, but Figure 3 shows that it was C. brevitarsis that fed on a human.  Please clarify the species.

Lines 323-325:  The information about how many blood-fed specimens were found and their percentages should be in Results.

Please state in Results how many mixed blood meals were found in individual specimens. A previous study of Culicoides in Thailand (Reference [32]:  Insects, 2022, 13(10), 912) found that the majority of specimens (almost 70%) had mixed-host blood meals in individual specimens.  Please discuss this major difference in the two studies.

Please check References very carefully to correct errors.  For example, in the first Reference, the first author name should be Borkent, A. not Borkent A.R.T.  Also, please correct numerous other errors.

Please read the manuscript carefully to correct grammatical mistakes and other language errors.  Discussion is sometimes hard to follow because of English language problems.  Probably best to have native English language person check the manuscript.

Author Response

Reviewer 2

Comments and Suggestions for Authors

This manuscript presents interesting results of sampling Culicoides around cattle pens in Thailand, including morphological and COI identifications and blood-meal analyses.  The following concerns should be addressed:

Please describe the sweep-netting procedure more clearly.  Sweep-netting should follow a standardized procedure if comparisons are to be made among locations. For example, the number of sweeps, height of the sweeps, and their configuration (typically a figure-8 motion) should be consistent for every sample. Otherwise, comparisons among sites becomes questionable.

Author response: Details of the sweep-netting procedure were added.

If 25 collections were made from 16 sites, how were the collections partitioned?  Clearly, some sites would have only one collection whereas other sites would have multiple collections.  Unless there was a standardized collecting procedure, comparisons among sites will be biased. For example, species richness would be expected to be proportionally greater where more collections were made.

Author response:  We considered that each collection represents sampling a site in data analysis. Therefore, all are representing by only one collection.

 If 12,916 midges were collected, how was the very small number of 79 specimens chosen for COI barcoding?

Author response:  We chose these specimens based on two criteria. First, specimens were chosen for COI study if more than one BINs were recorded in Barcode of Life Data Systems (BOLDs) (https://www.boldsystems.org/index.php) for the species. The COI barcoding sequences obtained from our specimens can then be  compared in BOLD to identify the BIN that they are belong to. Second, specimens were chosen for COI study if COI sequences of specimens from Thailand had not yet been reported. To clarify this, we added the criteria and reasons for specimen choice for COI barcoding study into the Materials and Methods section.

How were specimens selected for blood-meal analysis?  How many were analyzed?  (We do not learn the answer until line 251).  Were some of the same 79 specimens for COI barcoding used?  Was visible evidence of a blood meal used to determine which specimens would be used?  Please provide more explanation of all this.

Author response: We selected representative of all available blood-fed specimens of every species collected for host blood source identification. There are eight species for which blood-fed specimens were not found therefore, they were not included in this analysis. The blood-fed specimens of species that met our criteria for selection to use for COI barcoding analysis were also used for COI amplification. We have added this information to the Materials and Methods section.

Under section 2.2 Molecular Study, please separate the COI barcoding analysis and the blood-meal analysis into two separate paragraphs to avoid confusion.

Author response: Thank you for your suggestion, we have separated the COI barcoding analysis and host blood meal analysis into two separate paragraphs.

Table 1:  For the table, pleased define S in a footnote.  Footnote b (“Exact number of chicken…from cattle pens owner”) does not make sense.  Please correct the English.

Author response: Thank you for your suggestion. Definitions for abbreviations in Table have been added. The English language has been checked and corrected.

Figure 1.  Actually the figure shows much more than “Map of Thailand”, including other countries.  It shows a portion of Southeast Asia, including Thailand.

Author response:  Figure legend has been modified to make it consistent with the figure.

Figure 2 seems to be a mix of color and black and white photos. All photos should be black and white. Otherwise, the photos suggest that there is some significance to the different colors of wings.

Author response: Thank you for suggestion. This figure was modified to make it better quality.

Lines 264-265 state that C. brevipalpis fed on humans, but Figure 3 shows that it was C. brevitarsis that fed on a human.  Please clarify the species.

Author response: Thank you very much for the comment. The species that fed on humans is C. brevitarsis. We have corrected this point.

Lines 323-325:  The information about how many blood-fed specimens were found and their percentages should be in Results.

Author response: Thank you for your suggestion. We have moved this information to the Results section.

Please state in Results how many mixed blood meals were found in individual specimens. A previous study of Culicoides in Thailand (Reference [32]:  Insects, 2022, 13(10), 912) found that the majority of specimens (almost 70%) had mixed-host blood meals in individual specimens.  Please discuss this major difference in the two studies.

Author response:  Thank you for your comment and suggestion. Because in our study, we use the conventional PCR for amplification of vertebrate cyt b gene, we cannot identify the mixed-host blood meals in an individual specimen. For Ref. 32 (Sunantaraporn et al., 2022), they used conventional PCR for cyt b and multiplex PCR that allow them to identify the mixed-host blood in an individual specimen.

Please check References very carefully to correct errors.  For example, in the first Reference, the first author name should be Borkent, A. not Borkent A.R.T.  Also, please correct numerous other errors.

Author response: Thank you very much for suggestion, we have checked and corrected the errors.

Comments on the Quality of English Language

Please read the manuscript carefully to correct grammatical mistakes and other language errors.  Discussion is sometimes hard to follow because of English language problems.  Probably best to have native English language person check the manuscript.

Author response:  Thank you very much for your suggestion. The English language has checked again by native English person.

Reviewer 3 Report

The manuscript „ Diversity, abundance and host blood meal analysis of Culicoides Latreille (Dyptera: Ceratopogonidae) from cattle pens in different land use types from Thailand” is a comprehensive and well written study. Study describes abundance and diversity of Culicoides species among cattle pens in diverse habitats in Thailand. Molecular techniques were used for identification of midges, as well as for host blood meal identification. Obtained results are clearly explained and summarized in the provided Figures and Tables. There are just few comments:

Line 147 Figure legend of Figure 1: Map of Thailand indicates all together 16 sampling locations, not 15

Line 251_What was the basis for the selection of 141 female mites for further molecular identification of host blood source? In which location (land type) were these mites collected?

Are there any data available about occurrence of vector-borne infections among cattle in investigated locations?

Author Response

Reviewer 3

Comments and Suggestions for Authors

The manuscript „ Diversity, abundance and host blood meal analysis of Culicoides Latreille (Dyptera: Ceratopogonidae) from cattle pens in different land use types from Thailand” is a comprehensive and well written study. Study describes abundance and diversity of Culicoides species among cattle pens in diverse habitats in Thailand. Molecular techniques were used for identification of midges, as well as for host blood meal identification. Obtained results are clearly explained and summarized in the provided Figures and Tables. There are just few comments:

Line 147 Figure legend of Figure 1: Map of Thailand indicates all together 16 sampling locations, not 15

Author response: Thank you very much for suggestion, we have corrected this error.

Line 251_What was the basis for the selection of 141 female mites for further molecular identification of host blood source? In which location (land type) were these mites collected?

Author response:  We selected representatives of all available blood-fed specimens of every species collected for host blood source identification. There were eight species for which blood-fed specimens were not found. Therefore, they could not be included in this analysis. The sampling sites where specimen were used for blood meal identification have been added into Table 2.

 Are there any data available about occurrence of vector-borne infections among cattle in investigated locations?

Author response:  Data are available for occurrence of vector-borne disease in Thailand including the northeastern region of the country where almost all sample sites were located. This information has been added to the Introduction.